# Multidimensional helical dichroism from a chiral molecular nanoassembly

Yusheng Jin[1], Xinghao Wang[1], Zhijie Xia[1], Xiaoxu Rao[1], Xiaomei Chen [2], Kaixuan Li[2], Yucheng Jiang [2,3], Jiaru Chu[1], Dong Wu [1] ✉, Cheng-Wei Qiu [2,4,5] ✉ & Jincheng Ni [1,6] ✉

Detecting the chirality of molecules is of great importance in optics, biomedicine, and materials science. In chiroptical spectroscopy, it's crucial to achieve strong chiroptical signals with a minimal number of chiral molecules. The molecular chiroptical signals, however, are typically weak for chiral molecular sensing in conventional circular dichroism using photonic spin angular momentum, even in the presence of a large number of chiral molecules (micromoles to millimoles). Here, by involving chiral light-matter interaction with photonic orbital angular momentum, we demonstrate strong chiroptical responses that reflect the molecular chirality in a single chiral nanoassembly. We experimentally present the helical dichroism spectra of chiral nanoassemblies synthesized from L/D-cystines, consistent with electromagnetic simulations. The asymmetry factors in the fundamental wavelength and photoluminescence emission reach values of 0.53 and 1.18, respectively, exceeding those observed in the circular dichroism mechanism. To improve the dimensions of helical dichroism spectroscopy, we analyze helical dichroism in wavelength domain, polarization domain, and momentum space. Our findings not only expand the methods for trace chiral molecular sensing but also provide insights into chiral light-matter interactions.

Chirality, defined as an object's inability to be superimposed on its mirror image through translation or rotation, plays a crucial role in physics, biomedicine, and materials science[1-4]. Chiroptical spectroscopy is highly significant as enantiomers can produce significantly different responses when interacting with other chiral entities[5,6]. However, conventional chiroptical spectroscopy techniques that rely on the spin angular momentum (SAM) of light, such as circular dichroism (CD) spectroscopy, normally yield extremely weak chiroptical signals (an asymmetry factor of $10^{-5} \sim 10^{-3}$) due to the size mismatch between chiral molecules and the probing light[7-9]. Due to

the inherent nature of the phenomenon, detecting a substantial CD signal generally necessitates a large quantity of chiral molecules, typically in the range of micromoles to millimoles. Therefore, achieving strong chiroptical signals with a minimal number of chiral molecules, ideally even a single molecule, is crucial for ultrasensitive chiroptical spectroscopy.

In CD measurements, substantial efforts have been invested into strengthening chiral light–matter interactions by the size-matching principle. Advancements in nanophotonics enable the enhancement of CD signals as they focus light into nanometer dimensions, improving

[1]CAS Key Laboratory of Mechanical Behavior and Design of Materials, Department of Precision Machinery and Precision Instrumentation, University of Science and Technology of China, Hefei, China. [2]Department of Electrical and Computer Engineering, National University of Singapore, Singapore, Singapore. [3]Jiangsu Key Laboratory of Intelligent Optoelectronic Devices and Chips, School of Physical Science and Technology, Suzhou University of Science and Technology, Suzhou, China. [4]National University of Singapore Suzhou Research Institute, Suzhou, Jiangsu, China. [5]Department of Physics, National University of Singapore, Singapore, Singapore. [6]State Key Laboratory of Opto-Electronic Information Acquisition and Protection Technology, Anhui University, Hefei, Anhui, China. ✉e-mail: dongwu@ustc.edu.cn; chengwei.qiu@nus.edu.sg; njc@ustc.edu.cn

their sensitivity to surrounding molecules[10–24]. However, the molecular chiroptical signals using these artificial nanostructures are influenced by the intrinsic chirality of the artificial structures and the random distribution of chiral molecules. Recently, chiral molecules−involving self-assembly represent a simpler and more sensitive method for enhanced chiroptical signals (an asymmetry factor of $10^{-1}$, typically) by facilitating the transfer of chiral features from the molecular scale to the micrometer/nanometer scale[25–34]. Notably, the chirality of the molecules determines the chirality of the assemblies, allowing for an asymmetry factor as high as 0.42, which is derived from CD signals[33]. However, these chiral nanoassemblies, synthesized by the self-assembly of chiral molecules, adopt random orientations and distributions, which may limit the chiroptical signal and necessitate a large number of nanoassemblies (Fig. 1a, Supplementary Note 1). A potential remedy for these limitations can be achieved by arranging these nanoassemblies into periodic arrays[32], though this significantly increases the experimental complexity.

Optical orbital angular momentum (OAM) represents a promising candidate for chiral molecular sensing[35–43]. Owing to its helical wavefronts with local chiral characteristics and unbounded topological charges, vortex beams carrying photonic OAM have the potential to obtain a strong chiroptical signal and increase the versatility of

chiroptical spectroscopy. The OAM-dependent dichroism, also known as helical dichroism (HD), has been demonstrated in artificially engineered chiral microstructures and metasurfaces, achieving substantial asymmetry factors[39,40]. However, detecting a strong HD signal originating from chiral molecules remains a challenge. Although theoretical studies predict the occurrence of HD in chiral molecules[35,36,44–46], the underlying mechanism of vortex−molecule interactions is still poorly understood, as evidenced by the weak observed HD signals of <0.006[36]. To improve the HD signals of chiral molecules, nonlinear optical HD and hard X-rays HD have also been realized in chiral molecular sensing[47,48]. However, the experimental HD signals are still relatively small, usually ranging from 0.01 to 0.05.

Here, we propose an approach for strong HD detection by synthesizing single chiral nanoassemblies from chiral molecules (L/D-cystines). By translating the molecular-scale chiral characteristics to the micrometer scale that couples effectively with vortex light, we demonstrated a high asymmetry factor with single chiral nanoassemblies, which can reflect the molecular chirality. The asymmetry factors in the fundamental wavelength and photoluminescence spectrum reach 0.53 and 1.18, respectively, substantially surpassing those observed in circular dichroism mechanisms for chiral molecular sensing. Our findings not only expand the methods for trace chiral

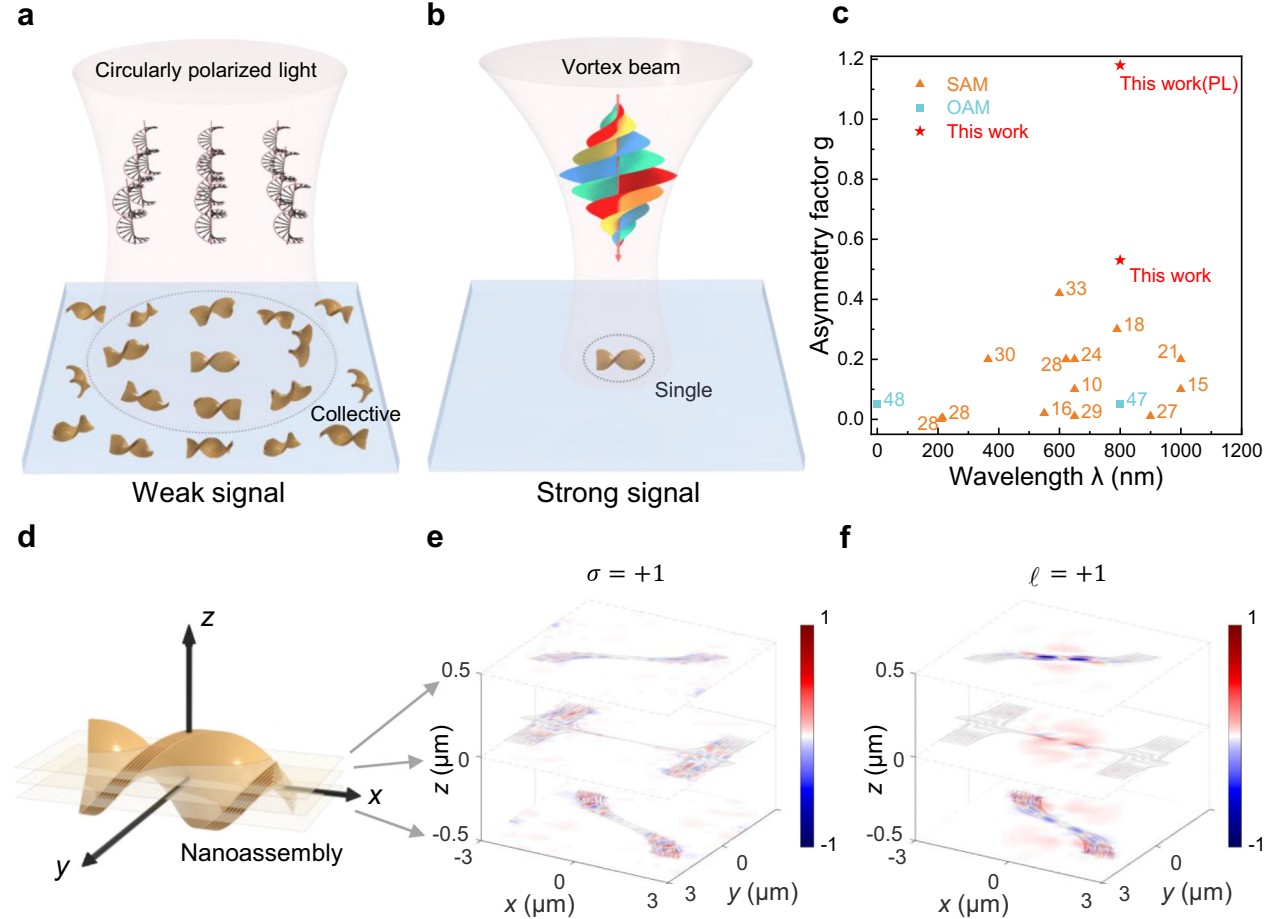

**Fig. 1 | Schematic illustration of chiroptical sensing using circularly polarized light versus vortex light. a** The chiral nanoassemblies synthesized by chiral molecules are typically randomly distributed on the substrate, resulting in a weak chiroptical signal by circularly polarized light. The chiroptical signal by circularly polarized light shows a collective property of chiral nanoassemblies. **b** A strong chiroptical signal can be achieved even on single nanoassemblies by vortex beams, yielding a large asymmetry factor from local interaction with optical OAM. **c** Asymmetry factors versus operating wavelengths of chiroptical sensing in chiral

molecules or nanoassemblies in this work and previous reports for chiral molecular sensing[10,15,16,18,21,24,27–30,33,47,48]. **d** Schematic of the chiral nanoassemblies. Local angular momentum flux density on nanoassemblies illuminated by circularly polarized light (**e**) and vortex beam (**f**) with the same total angular momentum, respectively. Colors represent the normalized values of the local angular momentum flux density (**e** and **f** are normalized by the same factor). The three simulated planes are indicated in (**d**). The shaded area represents the cross-section of the nanoassembly in that plane.

molecular sensing but also provide new insights into chiral light–matter interactions between vortex beams and chiral nanoassemblies.

## Results

### Chiroptical detection from optical SAM to OAM

A substantial chiroptical signal can be obtained in a single chiral molecular nanoassembly by HD measurements (Fig. 1b). When employing vortex beams for chiroptical detection, the local chiral properties are captured by focusing vortex beams to a size similar to the chiral nanoassembly (see Supplementary Figs. 2, 3). Distinct from CD signals from collective nanoassemblies, our HD measurements only require single chiral nanoassemblies.

We further studied the chiral light–matter interaction in nanoassemblies by simulating the local angular momentum flux densities[49]. As shown in Fig. 1d, the structural parameters are obtained from scanning electron microscopy (SEM) images of chiral nanoassemblies. The $z$-component angular-momentum flux density for the vortex beam through a transversal plane oriented in the $z$-direction can be separated as $M_{ZZ} = M_{spin} + M_{orbit}$. Specifically, the two angular momentum fluxes are given by $M_{spin} = \frac{1}{2\omega} Im[\mathbf{E}_x \mathbf{H}_x^* + \mathbf{E}_y \mathbf{H}_y^*]$; $M_{orbit} = \frac{1}{4\omega} Im[\mathbf{E}_y \frac{\partial \mathbf{H}_x^*}{\partial \varphi} - \mathbf{H}_x^* \frac{\partial \mathbf{E}_y}{\partial \varphi} + \mathbf{H}_y^* \frac{\partial \mathbf{E}_x}{\partial \varphi} - \mathbf{E}_x \frac{\partial \mathbf{H}_y^*}{\partial \varphi}]$ where $\omega$ is the angular frequency of the electromagnetic field[44]. A comparison of angular momentum flux density distributions is performed for a nanoassembly illuminated by circularly polarized light or vortex beams. To ensure consistent total angular momentum of the incident photons, circularly polarized light with $\sigma = +1$ and vortex light with topological charge $\ell = +1$ are used in the simulation, both with the same power. Under circularly polarized light illumination, the angular momentum flux density shows a sparse distribution with low values at various heights (Fig. 1e). Under vortex beam illumination, the angular momentum flux density is concentrated near the structure and exhibits a higher local density, indicating stronger chiroptical interaction with the nanoassembly (Fig. 1f). Local angular momentum flux densities for circularly polarized light and vortex beam are also simulated in free space, showing the different chiral-field distributions (see Supplementary Fig. 4).

We compared the asymmetry factors of chiroptical signals from chiral molecules in previous research and our work, as shown in Fig. 1c. Our approach achieves an asymmetry factor of 0.53 and 1.18 in the fundamental wavelength and photoluminescence emission, respectively, which represents a significant advancement in the identification and sensing of chiral molecules (see Supplementary Table S2).

### Experimental set-up and synthesis of chiral nanoassemblies

We transfer the chiral characteristics of chiral molecules by promoting their self-assembly into chiral nanoassemblies. L/D-cystine and cadmium chloride ($CdCl_2$) solutions were mixed in an alkaline condition to produce the chiral nanoassemblies. We then assessed the chiroptical HD signals of these nanoassemblies by incident linearly polarized vortex beams on the synthesized chiral nanoassemblies (Fig. 2a). Gaussian beams are converted into optical vortex beams with tunable OAM by displaying various holograms using a liquid-crystal spatial light modulator (SLM). By using a microscopic system, we have provided a quantitative analysis of the relationship between beam size and topological charge (see Supplementary Fig. 2), and adjusted the beam to ensure perpendicular incidence while precisely aligning it with the center of the nanoassembly. The HD measurement is schematically shown in Fig. 2b, illustrating the interaction between vortex beams and single chiral nanoassemblies. For vortex beams with opposite topological charges, the reflection intensity is different to deduce the chirality of the nanoassemblies. The nanoassemblies synthesized by L-cystine exhibit a right-handed helical arrangement, resulting in a larger reflection intensity for vortex beams with negative than positive topological charges (Fig. 2c).

The concentrations of the solutions and the relative proportions of the various components affect the size and chirality of chiral nanoassemblies[31] (see the "Methods" section for detailed synthesis). In our experiments, pure L-cystine-derived nanoassemblies have a right-handed helical shape, whereas pure D-cystine-derived nanoassemblies have a left-handed helical shape (Fig. 2d). We define the nanoassemblies formed from L-cystine (D-cystine) as L-Cys (D-Cys) nanoassemblies for simplicity. The SEM image of large-area D-Cys nanoassemblies exhibits uniform chirality and random distributions on the substrate (Fig. 2e).

By harnessing the ability to maintain chiral characteristics during the self-assembly of chiral molecules, we can effectively transfer the chiral properties of molecules from the molecular scale to the micrometer scale. This transition significantly enhances chiral light–matter interactions with vortex light in the visible spectrum, enabling more effective chiroptical detection of molecules. Furthermore, the chiral nanoassemblies investigated in our experiments reside in a regime of $D > \lambda$ for dimensional matching, where $D$ is the diameter of the nanoassemblies and $\lambda$ is the wavelength of visible light. This conceptual framework represents a significant departure from previous works[25–30,47,48], leveraging controlled self-assembly and the unique dimensional characteristics of vortex beams to achieve enhanced chiroptical performance.

### HD spectrum of chiral nanoassemblies

To mitigate fluctuations in the asymmetric signals caused by nearly zero reflection intensity in large topological charges, we define the HD asymmetry as:

$$HD = 2 * \frac{I^+ - I^-}{I_{max}^+ + I_{max}^-} \tag{1}$$

where $I^+$ and $I^-$ represent the reflection intensities under vortex beam illumination with topological charge $+\ell$ and $-\ell$, respectively. Meanwhile, the terms $I_{max}^+$ or $I_{max}^-$ denote the maximum reflection intensity obtained as the topological charge $\ell$ varies, effectively normalizing the differential signal.

The characteristic donut-shaped spatial distribution of the incident vortex beam with topological charge $\ell = +4$ is presented in Fig. 3a. The white arrow indicates both the magnitude (length) and direction (orientation) of the momentum density within the $x$–$y$ plane. Additionally, we numerically simulate the electric field distribution and reflection spectra for L/D-Cys nanoassemblies. We present the intensity distribution ($|\mathbf{E}|^2$) obtained from illuminating the D-Cys nanoassembly with a vortex beam ($\ell = +4$), as shown in Fig. 3b. This distribution is closely related to the chirality of the nanoassemblies, thereby confirming the presence of strong chiral light–matter interactions. Notably, the electric field distributions of vortex beams with opposite topological charges, initially identical, exhibit significant differences upon reflection from the nanoassemblies (see Supplementary Fig. 6). The intensity difference for the electromagnetic near-fields is consistent with the macroscopic asymmetric response of chiral nanoassemblies to vortex beams, which can be attributed to scattering at the nanoplanes that make up the nanoassemblies. The simulated chiroptical HD spectrum further corroborates these findings, displaying mirror images with respect to the zero line for the L-Cys and D-Cys nanoassemblies (Fig. 3c). For consistency, the geometric parameters of the chiral nanoassemblies used in the model are aligned with those employed in the experimental setup. In addition to pitch, length, and width, the segmented characteristics of the nanoassemblies are also considered, as they influence the chiroptical HD signal (see Supplementary Note 6). We have also conducted detailed simulations to assess the robustness of the chiroptical signals on beam-center alignment (see Supplementary Fig. 11).

The experimental normalized reflection spectra of vortex beams with opposite topological charges on the L-Cys and D-Cys

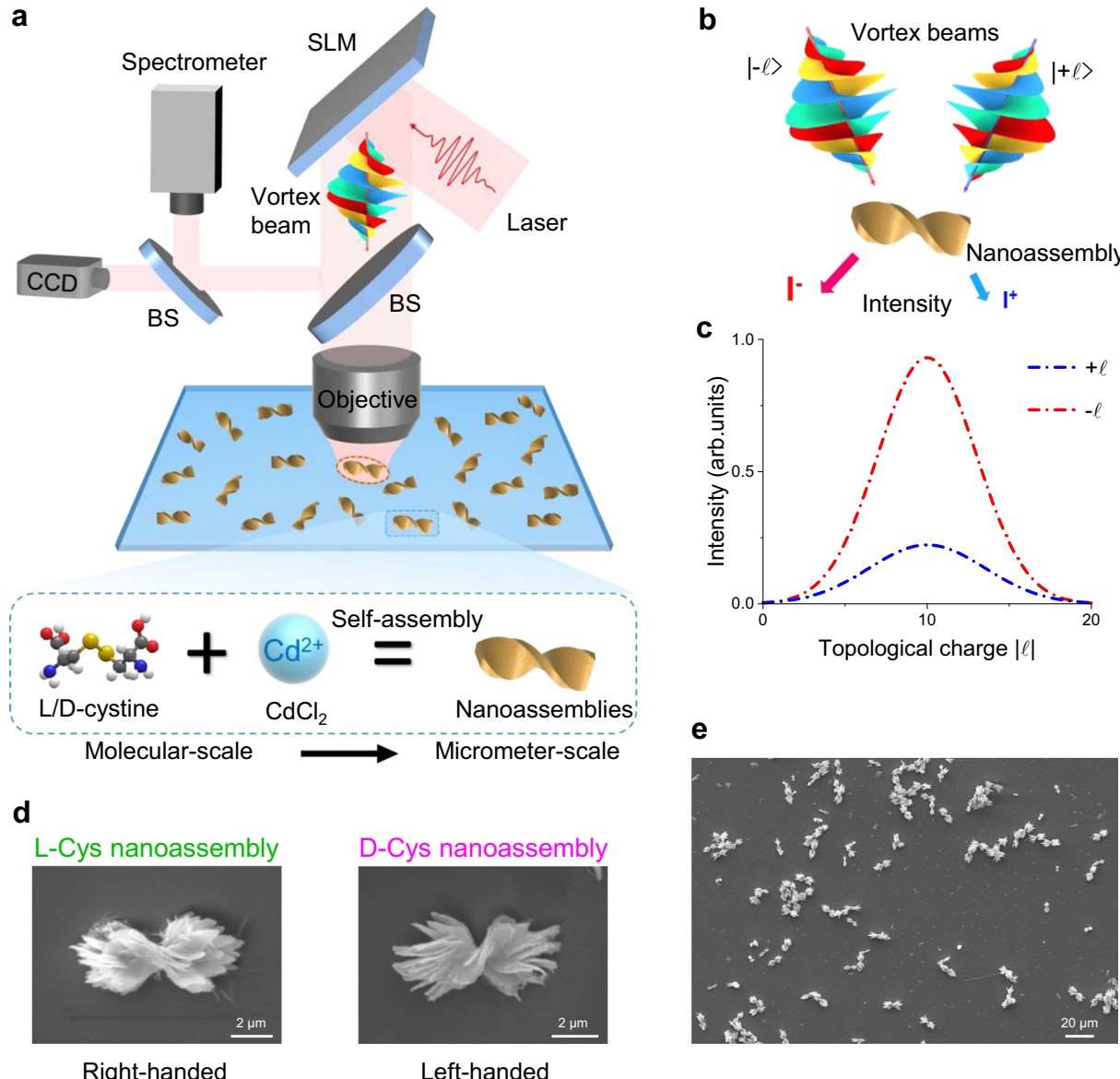

**Fig. 2 | Experimental characterization of chiroptical detection on chiral nanoassemblies synthesized by chiral molecules. a** Experimental setup of HD measurements on the synthesized chiral nanoassemblies. Linearly polarized vortex beams modulated by an SLM are focused on single chiral nanoassemblies for chiroptical detection. The chiral nanoassemblies are synthesized by self-assembly of chiral molecules from the molecule-scale to the micro-scale. BS: beam splitter.

**b** Schematic diagram of the chiroptical HD measurements on the chiral nanoassemblies by chiral light–matter interactions. **c** The chiroptical signal can be measured on the difference between reflection spectra by vortex beams with opposite topological charges of $\pm \ell$. **d** Enlarged SEM images of chiral nanoassemblies synthesized by ʟ-cystine (left panel) and ᴅ-cystine (right panel). **e** SEM image of the randomly distributed chiral nanoassemblies on the substrate.

nanoassemblies are depicted in Fig. 3d, e. For the ʟ-Cys nanoassembly, the reflection curves for vortex beams with positive and negative topological charges exhibit pronounced differences for charges ranging from $\ell = 0$ to 10. This discrepancy arises due to the strong chiral light–matter interaction between vortex beams and chiral nanoassemblies. Specifically, when the helicity of light aligns with that of the nanoassembly, strong coupling occurs, resulting in reduced reflection intensity. As the topological charge increases to $\ell = 30$, the curves converge closely, attributed to the increased diameter of the donut-shaped beam, which causes a mismatch that manifests as the reflection intensity approaching zero. For the ᴅ-Cys nanoassembly, the behavior is precisely the opposite (Fig. 3e). This observation is further validated by the experimental chiroptical HD spectra, which also demonstrate mirror images with respect to the zero line for the ʟ-Cys and ᴅ-Cys nanoassemblies (Fig. 3f). The

experimental results agree well with our simulations, though the lower peak in the simulated HD spectrum may be attributed to simplifications within the simulation model.

Notably, the chiroptical HD signal peaks at $|\ell| = 4$ for the ʟ-Cys nanoassembly, exceeding 0.5, representing a significant enhancement compared to other studies using optical OAM for chiral molecular sensing. To clarify the origin of HD, we conduct a multipolar decomposition analysis, and the results demonstrate that the electric quadrupole (EQ) component contributes significantly to the electromagnetic field response (see Supplementary Note 5). Moreover, while our experiments are conducted in reflection mode, this approach is also applicable in transmission mode. The HD signal in transmission mode exhibits an opposite sign but follows a similar trend to that in reflection mode, suggesting comparable chiroptical responses can be achieved (see Supplementary Fig. 7).

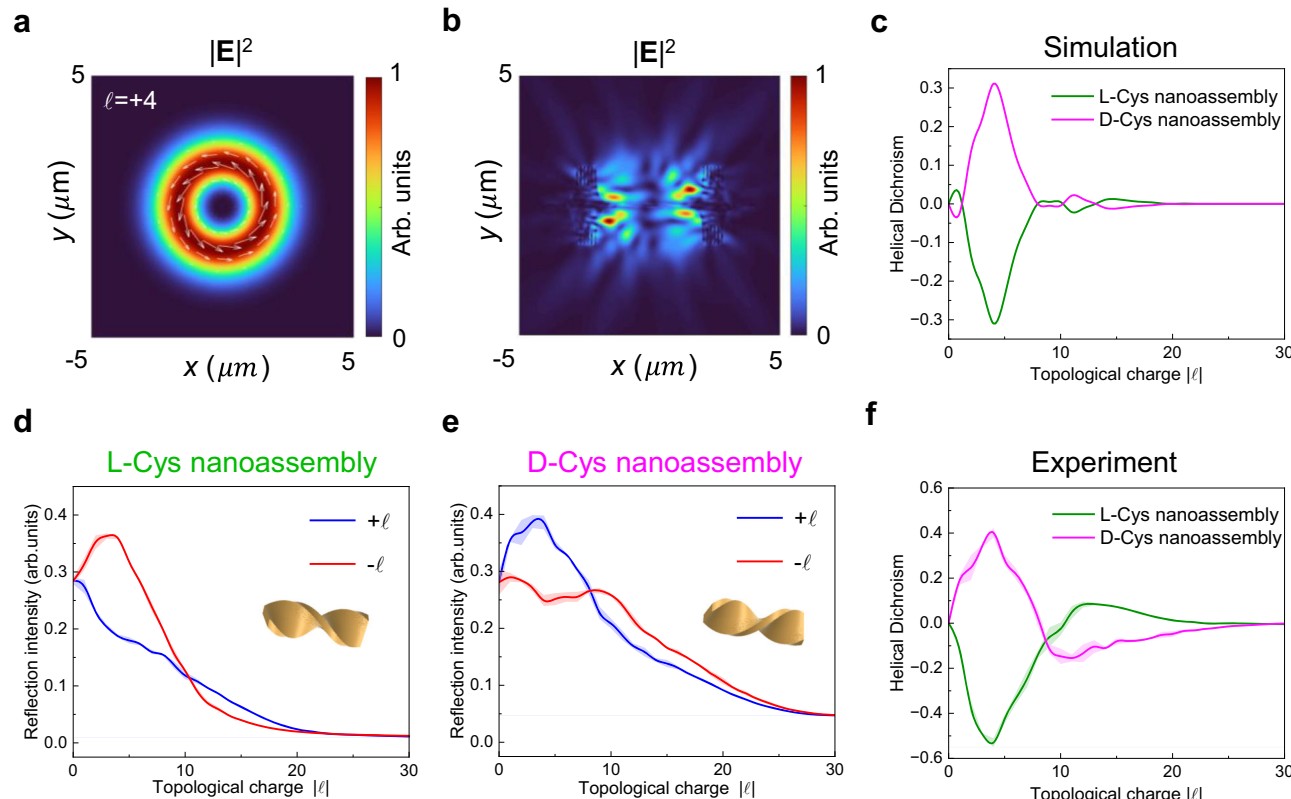

**Fig. 3 | Chiroptical HD spectra of chiral nanoassemblies. a** The simulated optical field distribution of the incident vortex beams with topological charge $\ell = +4$. The length and orientation of the white arrow are indicative of the magnitude and direction of the momentum density in the $x$–$y$ plane, respectively. **b** Simulated electric field intensity distribution of D-Cys nanoassembly illuminated by the vortex beam with topological charge $\ell = +4$ at the $z = 0$ plane. **c** Simulated chiroptical HD spectra on L/D-Cys nanoassemblies. Experimental reflection spectra of L-Cys (**d**) and D-Cys nanoassemblies (**e**) by vortex beams with topological charge from 0 to 30. The insets in **d** and **e** show schematic illustrations of the L/D-Cys nanoassembly. arb. units, arbitrary units. **f** Experimental chiroptical HD spectra on L/D-Cys nanoassemblies. Solid lines show the mean value, and the shading indicates the standard deviation of multiple measurements.

## HD spectrum of chiral nanoassemblies in photoluminescence

In addition to the reflection spectra, we measured the photoluminescence (PL) spectra of the synthesized nanoassemblies and realized a larger chiral asymmetry. The asymmetry factor in photoluminescence is defined as follows:

$$HD_{PL} = 2 * \frac{I_{PL}^+ - I_{PL}^-}{\max(I_{PL}^+) + \max(I_{PL}^-)} \quad (2)$$

where $I_{PL}^{\pm}$ represent the PL intensity.

Figure 4a illustrates the schematic of measuring $HD_{PL}$ by collecting the PL emission from chiral nanoassemblies. We are also able to capture the full fluorescence spectrum by replacing the charge-coupled device (CCD) camera with a spectrometer. Similar to the principles of HD spectra measurement, the chirality of the nanoassemblies can be detected by comparing the PL intensities under the illumination of vortex beams with opposite topological charges. We present the intensity difference ($\triangle|\mathbf{E}|^2$) on the D-Cys nanoassembly by subtracting the intensity between topological charge $\ell = +6$ and $-6$, where the predominance of positive values near the nanoassembly indicates stronger light-matter interaction with $\ell = +6$ (Fig. 4b). The $HD_{PL}$ originates from the absorption difference with opposite vortex beams as the strongest PL emission is generated by the strongest interaction between a specific handedness of OAM light and the nanoassembly (see Supplementary Fig. 14). Both L-Cys and D-Cys nanoassemblies exhibit chiral PL emission in the range of 500–700 nm under illumination with vortex beams (Fig. 4d, e). The dotted lines in the figures represent the raw experimental data, while the solid lines correspond to the fitted $HD_{PL}$ spectra. As expected, the corresponding

$HD_{PL}$ spectra of the L-Cys and D-Cys nanoassemblies display mirror symmetry with respect to the zero line (Fig. 4c). For the L-Cys nanoassembly, the asymmetry factor peaks at ~600 nm, showing a remarkable asymmetry factor exceeding 0.7. Furthermore, we observe that the asymmetry factor, derived from vortex beams incident on the D-Cys nanoassembly, varies with changes in the topological charge $|\ell|$ in the $HD_{PL}$ spectra (Fig. 4f). This variation follows a trend similar to that observed in the chiroptical HD spectra, but the asymmetry factor can reach a maximum value of 1.18 at the topological charge $\ell = 8$.

## HD spectrum of chiral nanoassemblies in wavelength domain, polarization domain and momentum space

To improve the dimensions of HD spectroscopy, we further explored the HD signals in the wavelength domain, polarization domain, and momentum space. Figure 5a presents a schematic diagram of the experimental setup, which utilizes a supercontinuum laser to vary the wavelength of the incident light over a tunable range of 450–650 nm. To enhance the accuracy of phase modulation during wavelength adjustments, the central wavelength of the SLM is synchronously tuned. The experimental HD signals at various wavelengths demonstrate that the topological charge associated with the HD peak value decreases as the wavelength increases, with peak values ranging between 0.2 and 0.4 (Fig. 5b). Additionally, we measured the CD signal on the same nanoassembly under identical tight focusing conditions (see Supplementary Fig. 13). The observed asymmetry factor for the CD signal is at the level of 0.05 by spin–orbit coupling, which indicates that interactions with photonic OAM can yield a larger asymmetry factor.

To investigate the impact of incident light polarization, a linear polarizer and a quarter-wave plate (QWP) are integrated into the

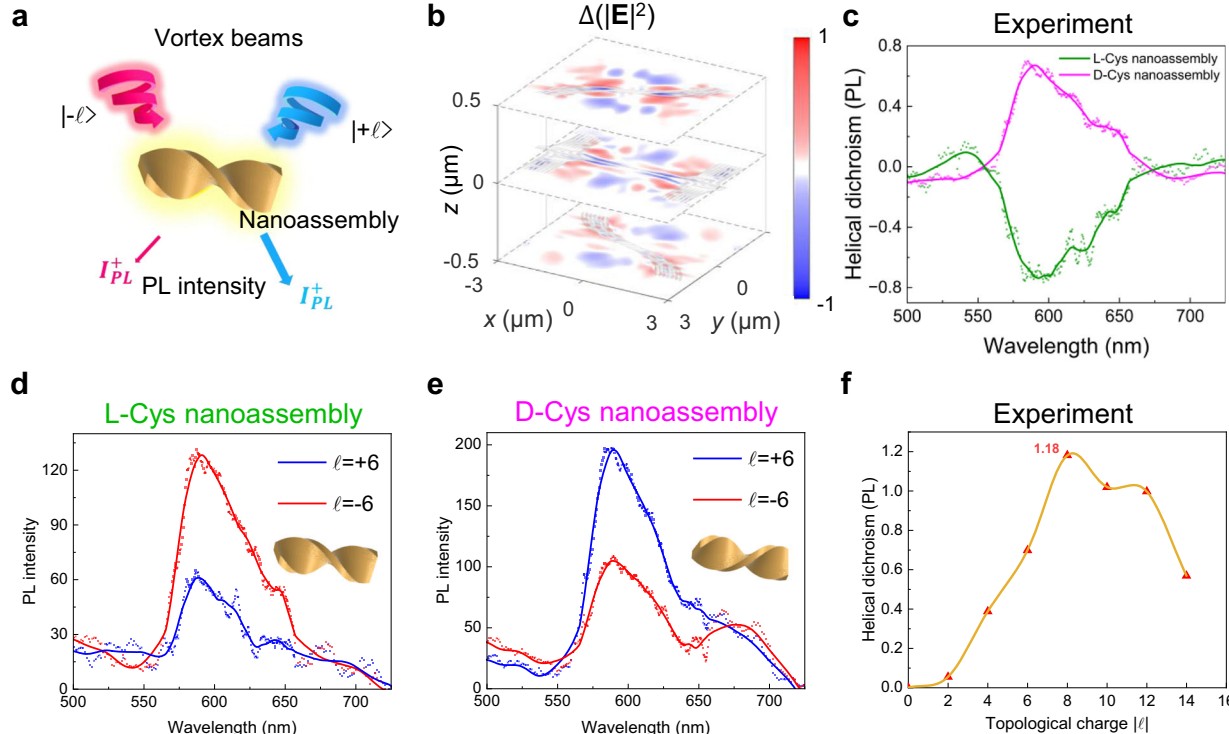

**Fig. 4 | Chiroptical HD spectra of chiral nanoassemblies in photoluminescence emission. a** Schematic of the PL excitation and collection. $I_{PL}^+$ and $I_{PL}^-$ denote the PL intensity with vortex beams carrying opposite OAM. **b** Simulated distributions of the electric-field difference $(\Delta(|E|^2) = |E|_{+\ell}^2 - |E|_{-\ell}^2)$ in the D-Cys nanoassembly illuminated by vortex beams with topological charges $\ell = \pm 6$. The shaded area represents the cross-section of the nanoassembly in that plane. Experimental PL spectra of L-Cys (**d**) and D-Cys nanoassemblies (**e**) by vortex beams with topological charges $\ell = \pm 6$ and corresponding HD$_{PL}$ spectra (**c**). The insets in **d** and **e** show schematic illustrations of the L/D-Cys nanoassembly. **f** Experimental HD$_{PL}$ spectra of the D-Cys nanoassembly by varying topological charges.

optical path, with the wavelength set to the commonly used 530 nm (Fig. 5c). The peak values of HD signal exhibit a sinusoidal variation with the rotation angle of QWP, reaching a maximum at 45° and a minimum at 135° (corresponding to circular polarization). This behavior stems from the conversion of partial SAM into OAM under tight-focusing conditions. Moreover, we examined the influence of the oblique angle on HD spectra. The results reveal that the peak value of the HD signal diminishes as the tilt angle increases, both in experiments and simulations (Fig. 5d). In addition to out-of-plane rotations, the impact of in-plane orientation relative to the light beam on the HD signal is also simulated, revealing that these rotations similarly reduce the peak value of the HD signal to different extents (see Supplementary Fig. 12). These findings highlight that the HD signal depends on the orientation of nanoassembly, which is crucial for optimizing the interaction between chiral nanoassemblies and photonic OAM.

## Discussion

By involving chiral light–matter interaction with optical OAM, we demonstrate strong chiroptical responses in single chiral nanoassemblies. This technique aims to exploit the preserved chiral characteristics of molecules during the self-assembly process, addressing the challenge of weak signal strength often encountered when directly detecting chiral molecules with vortex beams. The asymmetry factors in the fundamental wavelength and photoluminescence emission reach values of 0.53 and 1.18, respectively, exceeding those observed in the CD mechanism and underscoring the significant potential of optical OAM in chiral molecular sensing. We have also analyzed HD signals in the wavelength domain, polarization domain, and momentum space to improve the dimensions of HD spectroscopy.

In conclusion, we propose a method for chiral molecular sensing by detecting the chiroptical HD signal of chiral nanoassemblies synthesized from L/D-cystines. While the direct self-assembly of chiral molecules into nanoassemblies is specific to certain systems, the core principle of our approach—scaling the chirality of molecules from the molecular level to a dimension comparable to the wavelength of light for interaction with photonic OAM—holds broader potential. By amplifying the chiral dimensions of molecules, this concept can be adapted to other systems, such as in chiral molecule-mediated growth of gold nanoparticles or other self-assembly processes involving chiral molecules. Our method offers a paradigm of chiroptical detection for chiral molecules, presenting an exciting direction for future chiroptical research.

## Methods

### Synthesis of chiral nanoassemblies

L-cystine (Ultra-pure, 99.5%) and D-cystine (98%) were purchased from Macklin. Cadmium chloride (CdCl$_2$, 99%) and sodium hydroxide (NaOH, pellets, >96%) were purchased from Sinopharm Chemical Reagent Co., Ltd. The chiral nanoassemblies were synthesized by mixing solutions of L/D-cystine and cadmium chloride (CdCl$_2$) under alkaline conditions[31]. Stock solutions of L-cystine and D-cystine were prepared by dissolving appropriate amounts of each in 10 mL of deionized water to achieve concentrations of 2 mM. To aid the dissolution of cystine, 60 μL of 1 M NaOH solution was added to the cystine solutions, and the mixture was vigorously shaken until homogeneous. This solution was then mixed with an equal volume of a 2 mM CdCl$_2$ solution, ensuring thorough mixing by further shaking. Typically, 500 μL of the cystine solution was added to 500 μL of the CdCl$_2$ solution. This mixture was allowed to remain undisturbed at room temperature for 15 min to enable the complete self-assembly of the chiral nanoassemblies. For optical characterization, 1 μL of the suspension was pipetted onto a glass slide and allowed to air-dry. SEM images were obtained using a ZEISS EVO18 SEM at an accelerating

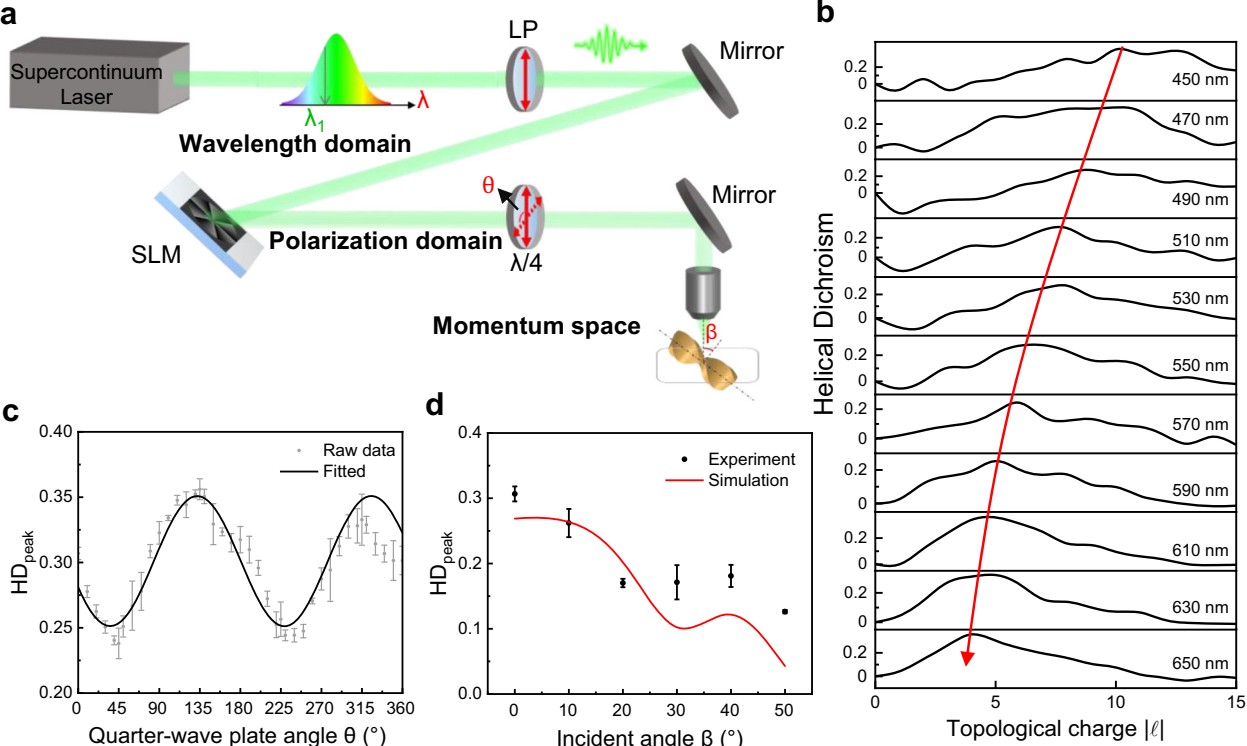

**Fig. 5 | Experimental chiroptical HD signals in wavelength domain, polarization domain, and momentum space. a** Schematic of the experimental setup. LP: linear polarizer; $\lambda/4$: quarter-wave plate. **b** Measured HD spectra at the varying operating wavelength from 450 to 650 nm. The red curve with an arrow is drawn as a guide to the eye, indicating the dependence of topological charge corresponding to the HD peak value. **c** Dependence of HD peak values on the quarter-wave plate angle $\theta$, measured at the laser wavelength of 530 nm. **d** Dependence of HD peak values on the incident angle $\beta$. $HD_{peak}$: peak value of the HD signal. The error bars represent the standard deviation of multiple measurements.

voltage of 10 kV, operating in secondary electron detection mode, after depositing ~10 nm of gold on the sample.

### Optical apparatus
A mode-locked Ti:sapphire ultrafast oscillator (Chameleon Vision-S, Coherent Inc.) is used as the femtosecond laser source, operating at a central wavelength of 800 nm with a pulse width of 75 fs and a repetition rate of 80 MHz. The polarization is controlled by a half-wave plate and a polarizer. A phase-only reflective liquid-crystal spatial light modulator (Pluto NIR-2, Holoeye Photonics AG) featuring a resolution of 1920 × 1080 pixels (pixel pitch: 8 μm) is utilized to display computer-generated holograms with varying topological charges. The vortex beams are focused onto the sample using an Olympus ×100 dry objective lens (NA = 0.9). A 3D piezo nanostage (E545, Physik Instrumente), with nanoscale resolution, is employed to enable precise adjustments of the sample position during optical microscopy. The reflection intensity distribution is captured using a CCD camera (MindVision HD-SUA133GM-T, image area: 1280 × 1024 pixels), with an exposure time of 10 ms. The PL intensity is measured using a spectrometer (MX Pro, OceanOptics). A supercontinuum laser (NKT Photonics EU-15) generated a laser beam with a variable wavelength. The modulation range of the spatial light modulator (X15213, Hamamatsu) corresponding to this laser is 400–700 nm.

### Numerical simulation
Electromagnetic numerical simulations are conducted using a full-wave finite-difference time-domain method, from which the electric field distribution and the simulated HD spectrum are generated. The simulation domain is set to 10 × 10 × 5 μm³, chosen to ensure accurate representation of the chiral microstructure and sufficient space to capture the electromagnetic field. Perfectly matched layer boundaries

are applied in the $x$, $y$, and $z$ directions, and the minimum mesh step is set to be 0.25 nm. The model of the chiral nanoassembly, composed of seven pieces, is consistent with the actual dimensions derived from the SEM images. The refractive index of chiral nanoassemblies is set to 2.5 (see Supplementary Note 4). The linearly polarized vortex light source with an operating wavelength of 800 nm is positioned above the structure. The electrical field distribution $E_{source}$ in polar coordinate space can be described as

$$E_{source} = C \frac{r^{|l|}}{\sqrt{|l|!}} e^{-\frac{r^2}{w_0^2}} e^{il\varphi} e^{-\frac{ikr^2}{2f}} \qquad (3)$$

where $C$ is a normalized constant independent of $l$ and $r$, $w_0$ is the beam waist, $k$ denotes the wave vector, and $f$ is the focal length. The incident vortex light is focused on the center of the nanoassembly by superimposing a spherical wave phase.

## Data availability
The main data supporting the results in this study are available within the paper and its Supplementary Information. Other source data that support the findings of this study are available from the corresponding authors upon request.

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

## Acknowledgements

This work was supported by the National Natural Science Foundation of China (Grant Nos. 62375253 and 62325507), National Key R&D Program of China (2021YFF0502700 and 2023YFF0613600). C.-W.Q. acknowledged the financial support by the Ministry of Education, Republic of Singapore (Grant Nos. A-8002152-00-00 and A-8002458-00-00), and the Competitive Research Program Award (NRF-CRP26-2021-0004 and NRF-CRP30-2023-0003) from the National Research Foundation, Prime Minister's Office, Singapore. C.W.Q. also acknowledged the support from the National University of Singapore Suzhou Research Institute via Grant No. R-2023-S-011. We acknowledge the Experimental Center of

Engineering and Material Sciences at USTC for the fabrication and measurement of samples. This work was partly carried out at the USTC Center for Micro and Nanoscale Research and Fabrication.

## Author contributions

Y. Jin and J.N. proposed the idea and conceived the experiment. Y. Jin, X.W., and Z.X. performed the experiments. Y. Jin, J.N., and X.R. performed the data analysis. Y. Jin and J.N. performed the numerical simulations. Y. Jin, J.N., and C.W.Q. wrote the manuscript. X.C., K.L., Y. Jiang, and J.C. revised the manuscript. J.N., D.W., and C.W.Q. supervised the project.

## Competing interests

The authors declare no competing interests.
