## [Transparent Peer Review file · Nature Communications]

Multidimensional helical dichroism from a chiral molecular nanoassembly

Corresponding Author: Professor Jincheng Ni

Version 0:

Reviewer comments:

Reviewer #1

(Remarks to the Author)

I am satisfied with the author response to all the reviewers and changes made to the manuscript. I recommend it for publication in Nature communications.

Reviewer #2

(Remarks to the Author)

all concerns raised by this reviewer have been addressed in this new version of the manuscript

**Responses to reviewers' comments (Manuscript number NCOMMS-25-90773
entitled "Multidimensional helical dichroism from a chiral molecular
nanoassembly" submitted to Nature Communications)**

We would like to thank all the reviewers for their careful reviewing of our work and their valuable comments/suggestions. These constructive comments/suggestions have allowed us to significantly improve the manuscript.

Reviewer #1 (Remarks to the Author):

I am satisfied with the author response to all the reviewers and changes made to the manuscript. I recommend it for publication in Nature communications.

Reply: We thank Referee #1 for his/her careful review, and the final suggestion of acceptance.

Reviewer #2 (Remarks to the Author):

All concerns raised by this reviewer have been addressed in this new version of the manuscript.

Reply: We thank Referee #2 for his/her help with the review of our manuscript.